# Carbon Nanodots Embedded on a Polyethersulfone Membrane for Cadmium(II) Removal from Water

**DOI:** 10.3390/membranes11020114

**Published:** 2021-02-05

**Authors:** Simanye Sam, Soraya Phumzile Malinga, Nonhlangabezo Mabuba

**Affiliations:** Department of Chemical Sciences (Formerly Known as Applied Chemistry), Doornfontein Campus, University of Johannesburg, P.O. Box 17011, Johannesburg 2028, South Africa; simanyesam@gmail.com (S.S.); smalinga@uj.ac.za (S.P.M.)

**Keywords:** cadmium(II), polyethersulfone, membrane, carbon nanodots, electrochemical sensing

## Abstract

Cadmium(II) is a toxic heavy metal in aquatic systems. As a potential solution, green carbon nanodots (CNDs) were synthesized from oats and embedded on polyethersulfone membrane (PES) via phase inversion for the adsorption of Cd^2+^ from water. Characterization techniques for the CNDs and PES membranes were transmission electron microscopy (TEM), Fourier transform infrared spectroscopy (FTIR), scanning electron microscopy (SEM), Raman spectroscopy, atomic force microscopy (AFM), contact angle and a pure water flux assessment system operated at 300 kPa. TEM results showed that the CNDs were well dispersed with a uniform shape and size (6.7 ± 2.8 nm). Raman spectroscopy revealed that the CNDs were embedded on the PES and the I_D_/I_G_ ratio slightly increased, showing that the membranes maintained good structural integrity.The CNDs/PES proved to be more hydrophilic than PES. The glassy carbon electrode (GCE) in anodic stripping voltammetry (ASV) technique detected 99.78% Cd^2+^ removal by 0.5% CNDs/PES at optimum conditions: 30 min. contact time, at pH 5 and 0.5 ppm Cd^2+^ solution. The 0.5% CNDs/PES removed Cd(II) due to the hydroxyl group (-OH) and carboxyl group (-COO-) on the membrane composite. It was established that Cu^2+^ and Pb^2+^ have a significant interfering effect during the analysis of Cd^2+^ using GCE in ASV technique. The 0.5% CNDs/PES is recyclable because it removed above 95% of cd^2+^ in four cycles. In a spiked tap water sample, 58.38% of Cd^2+^ was sensed by GCE of which 95% was in agreement with the value obtained from inductively coupled plasma optical emission spectrometry (ICPOES).

## 1. Introduction

Water contamination due to heavy metals has become a worldwide concern since the 1990s and has been a challenge for many environmental scientists [1]. The contribution of metal contamination in aquatic environments is caused by the colossal quantity of toxic heavy metals released into these environments by anthropogenic activities as well as by natural processes. These high concentrations of heavy metals are found mostly in sediments rather than in the water columns because they tend to accumulate in bottom deposits because of their higher density compared to water [2]. In portable water, wastewater and environmental water, poisonous heavy metals such as arsenic, cadmium, chromium, copper, lead and mercury are encountered [3,4,5,6,7]. Cadmium (II) is one of the most commonly encountered toxic heavy metals in water. It is demonstrated that +2 oxidation state is the abundant and most toxic of its compounds although it also exists in +1 state [8]. Cadmium (II) has been widely dispersed in the environment through industrial processes such as printed board manufacturing, metal finishing, plating, textile dyes, manufacturing of nickel-cadmium batteries, mining and smelting as stated by Malecki and Maron [9]. It is also distributed through human activities such as utilization of compost and discarding of Ni-Cd batteries where the cadmium is deposited into the rivers by rain as mentioned by Fatoki and Awofolu [10].

Cadmium(II) can bioaccumulate in seafood and plants, which is of grave health concern as some plants and seafood are consumed by humans and animals. Human exposure to Cd^2+^ can lead to many types of diseases such as nephrotoxicity, kidney disease, renal function hypertension, hepatic injury, lung damage, teratogenetic effects, skeletal deformation (Itai-itai) and cardiovascular diseases [11]. Cd^2+^ is also a carcinogen [1,2,11,12]. South Africa has had some disputes concerning the Cd^2+^ levels in river water. The normal concentration of Cd^2+^ in freshwater should be 0.005 mg/L.

In South Africa, the range guideline is 0 to 0.005 mg/L in river water for domestic use according to Department of Water Affairs and Forestry (DWAF) [13]. Water samples collected from South African water systems such as the Tyume River, Buffalo River, Keiskamma River, Umtata River and Sandile Dam, were found to have elevated concentrations of Cd^2+^. The concentrations ranged from 0.003 to 0.044 mg/L with the Tyume River having the highest Cd^2+^ concentrations of 0.030 ± 0.002 to 0.044 ± 0.003 mg/L. This is mainly due to the water runoffs from agricultural soils where phosphate fertilizers were used. Cd^2+^ is a common impurity in phosphate fertilisers [9,11,14,15]. Chemical precipitation, ion exchange, reverse osmosis, phytoremediation and adsorption are methods applied in heavy metal removal in water [3,4,5,6,7,16,17]. Adsorption is the most convenient and widely researched technique because of its ease of use and low cost [4]. Materials such as activated carbon/zicornium oxide composite activated carbon produced from rubber wood sawdust, wastewater sludge, cerium dioxide and its composites [3,4,6,7]. Phytoremediation is a green method that is highly promising in the alleviation of heavy metals in the environment and it is required to be more explored [5]. All the abovementioned methods are crucial for water treatment. However, their application is limited due to the a number of drawbacks: high operating costs, high costs of disposal of the precipitated sludge that is formed, performance affected by long waiting periods for plants to grow in phytoremediation, lack of control of the adsorbent in water and generally high maintenance costs due to fouling which necessitates cleaning [17]. The innovative membrane technology offers an avenue for exploration in the removal of cadmium in water and has successfully been applied in water treatment [18,19]. The polyethersulfone (PES) membrane possesses good chemical resistance, has an outstanding oxidative stability, has wide pH and temperature tolerance and mechanical strength [20]. However, membrane application in water treatment is limited due to its low hydrophilic nature and fouling [21]. Therefore, the novel aspect of our work is to improve the characteristics of PES by coating it with carbon nanodots (CNDs) which seeks to monitor and adsorb cadmium(II) from water and industrial effluents. The composite (CNDs and PES) will be reusable. A green method of synthesizing the CNDs from oats (organic cereal) was applied to produce carbon nanodots containing hydroxyl (OH-) and carboxylate (COO-) functional groups, which assisted in increasing the hydrophilicity of PES [13]. The carbon nanoparticles also have fluorescence properties, good biocompatibility and low toxicity for water treatment [13,22]. Therefore, CNDs will be embedded on a microporous membrane made of polyethersulfone (PES) to increase the membrane hydrophilicity and the removal of Cd^2+^ via adsorption, since they are good natural adsorbents.

## 2. Materials and Methods

### 2.1. Synthesis of Carbon Nanodots and Membrane

Commonly consumed whole grain oats (Jungle Oats) produced in South Africa was purchased from the local supermarket. Polyethesulfone (PES), polyvinylpyrrollidine (PVP) and 1-methyl-2-pyrolidinone (NMP) were purchased from Sigma Aldrich (St. Louis, MO, USA). The glassy carbon electrode (GCE), reference electrode and platinum auxiliary electrode were purchased from BASi (Newport Beach, CA, USA).

Carbon nanodots (CNDs) were synthesized according to a method reported by Shi et al. with modification. Oats (20 g) were placed in a crucible, transferred into a muffle furnace and pyrolyzed at 400 °C for 2 h instead of being microwaved [13]. The black product was cooled to room temperature and then mechanically crushed to a fine powder. The powder was then dispersed in ultrapure water and centrifuged several times to remove larger particles. The carbon nanodots aqueous suspension was filtered and the CND powder was obtained after drying in an oven at 80 °C for 24 h.

Phase inversion via immersion precipitation was used to synthesize the membranes. This method allowed for the change in phase of materials from the liquid to solid phase. Different amounts of CNDs were mixed with 1-methyl-2-pyrrolidone (NMP) and sonicated for 15 min to encourage dispersion (Table 1). PES (16 g) and polyvinylpyrrolidine (2 g) were added to the CNDs mixture and stirred for 24 h to prepare the casting solution. A casting knife (Elcometer 3545 Adjustable BirdFilm Applicator, Claremont, South Africa) was used to cast 150 µm membranes on a glass plate. After casting, the solutions were immediately submerged in a coagulation bath containing water (non-solvent). The membranes were further submerged into another water bath for 24 h to ensure that it was free from NMP. The membranes were then air dried for 24 h and sandwiched between plain sheets of paper for storage. The membrane composition was varied according to the percentage of CNDs added, as displayed in Table 1.

### 2.2. Characterisation of Membrane Embedded with Carbon Nanodots

The morphology and size distribution of the CNDs was characterised by TEM using JEOL JEM-2100 (Pleasanton, CA, USA). Pristine CNDs, pristine PES, 0.01% CNDs/PES, 0.05% CNDs/PES and 0.5% CNDs/PES were analysed using a Fourier Transform Infrared (FTIR) (Perkin Elmer, Spectrum 100, Shelton, CT, USA). Raman spectroscopy was used for surface characterization of the partially ordered carbon nanodots. The Raman spectrometer Perkin Elmer, Raman Micro 200, Waltham, MA, USA, was used with an output laser power of 50%. The spectra were recorded over a range of 50–3270 cm^−1^ using a spectral resolution of 2.0 cm^−1^. The hydrophilicity of the membranes was analysed using a sessile drop method on a Data Physics optical contact angle instrument (SCA 20 software, Camberley, Surrey, UK).

Scanning electron microscopy (SEM) was used to study the surface morphology and cross-sectional images of the membranes. SEM images were analysed at an accelerating voltage of 2 kV using a TESCAN Vega TC instrument (VEGA 3 TESCAN software, Brno, Czech Republic). The SEM instrument was equipped with an X-ray detector for energy dispersive X-ray analysis (EDX), which was operated at 5 kV. The membrane’s topological properties and roughness (Rq) were analysed using a Veeco Dimension 3100 atomic force microscope (AFM) equipped with V530r3sr3 software (Plainview, NY, USA) in 3D mode at 5 µm scan. The tip was mounted onto a 225 µm cantilever with a spring constant of 2.8 N/m.

### 2.3. Evaluation of the Membrane Performance

Sterlitech (Kent, WA, USA) dead-end filtration system was used to evaluate the pure water flux of the pristine and modified membranes. The membranes were first compacted at 300 kPa for 15 min for stabilization. Six different pressures were used for flux measurements from 300 kPa, 250 kPa, 200kPa, 150 kPa, 100kPa and 50 kPa and the flux calculated using Equation (1).
(1)$$j=\frac{v}{A\Delta t}$$
where *j* is the water flux (L/m^2^ h), *v* is the permeate volume (L), *A* is the membrane area (0.00146 m^2^) and Δ*t* is the change in filtration time (h).

### 2.4. Adsorption of Cadmium(II) from Water Using PES Membrane Coated with Carbon Nanodots

A method adopted from a study performed by Zhu et al. with modification was used to carry out batch adsorption experiments [23]. PES membrane (16 cm^2^) was immersed in 25 mL of Cd^2+^ synthetic solutions and shaken at different time intervals (1, 5, 15, 30, 60 and 120 min). The supernatants were then collected and analysed using Anodic stripping voltammetry (ASV) and inductively coupled plasma optical emission spectroscopy (ICP-OES). The effect of pH, contact time, standard concentration and carbon nanodots concentration on membranes were investigated using ASV measurements. The amount of Cd^2+^ adsorbed was calculated using Equation (2).
(2)$$R\;\left(\%\right)=\;\frac{C_{i}-C_{f}\;}{C_{i}}\;\times 100$$
where *C_i_* (mg/L) is the initial concentration of the metal ions in aqueous solution, *C_f_* (mg/L) the final concentration of the metal ions in solution. The adsorption capacity of the system was also calculated using Equation (3).
(3)$$Adsorption\;capacity=\;\frac{C_{i}-C_{f}}{A}\;\times V$$
where *C_i_* is the initial concentration of Cd^2+^, *C_f_* is the Cd^2+^ concentration after adsorption (mg/L), *A* is the membrane area (cm^2^) and *V* is the volume of the Cd^2+^ solution (L).

### 2.5. Electrochemical Detection of the Cadmium(II) Removed from Water

Square wave anodic stripping voltammetry (SWASV) was used to detect Cd^2+^ in 0.1 M HCl on bare glassy carbon electrode (GCE). The deposition potential and time were −900 mV and 200 s respectively. The stripping of cadmium was achieved by scanning at a potential range between 0.6 V and 2 V. The calibration studies were explored by using cadmium(II) standard solutions with concentrations ranging from 0.1 to 10 ppm under optimised conditions. In each calibration point, three replicates of SWASV measurements (n = 3) were done in order to minimise errors. The peak current signal increased proportionally with the increase in concentration. The linear regression equation obtained for the Cd^2+^ detection was found to be Y = 1.4275 × 10^−4^ + 4.344906 × 10^−5^ with a correlation coefficient (R^2^) of 0.98489. The limit of detection (LOD) was determined by three times multiplication of the standard deviation of the blank and division by slope of the calibration curve (3σblank/slope). The limit of quantification was (LOQ) which reflected the accepted measurements at the lowest concentration was calculated as 10σblank/slope. The calculated LOD and LOQ were as 0.0014 and 0.0046 ppm respectively. The bare GCE electrode was applied in the detection of cadmium in real water sample and the results were validated by inductively coupled plasma optical emission spectroscopy.

## 3. Results

### 3.1. Characterisation of the Membrane

The TEM images of the carbon nanodots are shown in Figure 1a. The TEM micrograph illustrated a mean particle size of 36.9 ± 11.0 nm for the carbon nanomaterials before it was mechanically ground to a fine powder. Well-dispersed and spherical carbon nanodots with uniform shape and size in the range of 2 to 10 nm with an average of 6.7 ± 2.8 nm were attained after mechanical grinding (Figure 1b). Similar results were observed by Shi et al., who synthesised duel emission carbon nanodots from naked oats via pyrolysis and mechanical grinding. In their study the majority of the carbon nanodots were in the range of 7 to 11 nm with an average mean of 8.64 ± 0.84 nm [13]. After the addition of CNDs, the surface of the membrane becomes more porous with fewer macrovoids. Consequently, the surface area was increased and more adsorption sites will be available for Cd^2+^ adsorption.

Fourier transform infrared spectroscopy (FTIR) analysis of the pure oats and CNDs is displayed in Figure 2a The pure oats had characteristic peaks at 3441 cm^−1^, 2920 cm^−1^, 1628 cm^−1^, 1399 cm^−1^ and 1147 cm^−1^ ascribed to -O-H, -COO^−^, C=C and -C-O-stretch respectively. Similar peaks were observed for the CNDs as illustrated in Figure 2a. The CNDs characteristic peaks were detected at 3441 cm^−1^, 2920 cm^−1^, 1628 cm^−1^, 1399 cm^−1^ and 1147 cm^−1^ ascribed to -O-H, C-H, COO-, C=C, and -C-O stretching vibrations respectively. This trend was also observed by Shi et al. In their study, similar observations of peaks 3426 cm^−1^, 3151 cm^−1^, 1634 cm^−1^ and 1400 cm^−1^, 1114 cm^−1^ and 1165 cm^−1^, 952 cm^−1^ were attributed to the O-H, C-H (stretching vibrations), COO-, C-O and O-H (bending vibrations) respectively.

Structural modification of PES using CNDs was also determined using FTIR in Figure 2b. The characteristic FTIR analysis and the respective peaks for pristine PES were observed at 1600 cm^−1^ and 1400 cm^−1^ for aromatic skeletal vibrations, 1324 cm^−1^ and 1239 cm^−1^ for C-O-C stretching and 1151 cm^−1^ and 1105 cm^−1^ for pristine PES membranes. The results are in agreement with other reports [24]. Blending the PES with CNDs introduced a new functionality at 3400 cm^−1^ which was attributed to -OH due to the presence of CNDs. The PES peaks observed between 1672 cm^−1^ to 500 cm^−1^ could mask the absent peaks from CNDs. Further analysis in the following sections, however, clarify the presence of CNDs within the membranes.

Figure 3 displays the Raman spectra for pure CNDs, 0.01% CNDs/PES, 0.05% CNDs/PES and 0.5% CNDs/PES membranes. The pure CNDs spectra showed two peaks which are generally attributed to the D-band at approximately 1339.02 cm^−1^ and G-band at ~1567.64 cm^−1^ [25]. For the modified membranes, these analyses generally showed the G and D band at ~1337.58 cm^−1^ and ~1592.69 cm^−1^ respectively. The G band is attributed to intrinsic vibrations of sp^2^ bonded graphitic carbon, whilst the D band corresponds to defects induced in the CNDs due to the disruption of -C=C bonds [26]. The G band was higher than the D band, showing good structural integrity of the CNDs. After blending the PES with CNDs, the I_D_/I_G_ ratio slightly increased to 0.74, 0.79, 0.83 and 0.82 for Pure CNDs, 0.01% CNDs/PES, 0.05% CNDs/PES and 0.5% CNDs/PES respectively as demonstrated in Table 2, the addition of the CNDs to the pure PES increased the I_D_/I_G_ ratio, which means that the membrane maintained good structural integrity as it was increasing to closer to one [27].

Figure 4a–d illustrates scanning electron microscopy (SEM) images of the surface and cross-section of PES and CNDs/PES membranes. The surface of PES is homogeneous and smooth without obvious voids or defects (Figure 4a) in comparison to the CNDs/PES membranes (Figure 4b–d). The CNDs/PES membranes presented a relatively spongy and porous surface as the amount of the CNDs was increased. The pure PES membrane showed a uniform thin finger-like structure. The tear-shaped elongated micro voids were observed extending towards the permeate side of the membrane (Figure 4a). As the CNDs concentration was increased, the macro voids within the membranes became more pronounced. In 0.01% CNDs/PES (Figure 4b), 0.05% CNDs/PES (Figure 4c) and 0.5% CNDs/PES (Figure 4d) membranes, a more pronounced cross-sectional asymmetry was observed. The elongated narrower macro voids of pure PES transitioned into wider macro voids that spanned the entire cross-section of the membrane. Orooji et al. observed a similar trend in their study of nano-structured carbon polyethersulfone composite ultrafiltration membrane with significantly low protein adsorption and bacterial adsorption adhesion [28]. In this study it was observed that the increase in viscosity due to addition of the mesoporous carbon leads to a slower solvent exchange process and eventually formed larger finger-like pores [28].

Atomic force microscopy (AFM) analysis was investigated out in 3D mode at 5 µm to observe the changes in surface topography as displayed in Figure 5. In Figure 5a, it was observed that the surface of the pristine PES was smooth compared to the CND embedded membranes. The rougher surface of the embedded membranes was created by the addition of the CNDs to the pure PES (Figure 1b–d). The CND embedded membranes also showed a varied ridge-and-valley structure as compared to the more uniform ridge-and-valley structure of the pristine PES. The roughness measurements (R_q_) in Table 3 were found to confirm the aforementioned results. The roughness measurements were 16.4, 21.9, 23.7 and 35.9 nm for pristine PES, 0.01% CNDs/PES (Figure 5b), 0.05% CNDs/PES (Figure 5c) and 0.5% CNDs/PES (Figure 5d), respectively. Therefore, the surface roughness increased with the increased amount of CNDs embedded to the membranes. Yuan et al. also reported that the incorporation of CNDs into their polyethyleneimine (PEI) matrix, dip-coated on polyacrylonitrile support, led to an increase in surface roughness [29].

The water contact angle analysis in this study was performed using the sessile drop method. This was done to investigate the hydrophilic nature of the membranes. Figure 6 shows the contact angle analysis of the pristine PES, 0.01% CNDs/PES, 0.05% CNDs/PES and 0.5% CNDs/PES modified membranes. The contact angle of pristine PES, 0.01% CNDs/PES, 0.05% CNDs/PES and 0.5% CNDs/PES were 73.4° ± 2.5°, 68.2° ± 7.8°, 64.8° ± 3.1° and 60.5° ± 3.7° respectively. The increased amount of CNDs reduced the contact angle, i.e., it improved the hydrophilicity of the membranes. This enhancement in hydrophilicity was attributed to the presence of hydrophilic functional groups such as -OH and COO^−^ found in the CNDs as reported in the FTIR analysis. A similar trend was observed by Orooji et al. [29] in whose study it was witnessed that, due to the added carbonyl functional groups of the mesoporous carbon, the contact angle of the control PES decreased, which meant that the membrane became more hydrophilic.

### 3.2. Membrane Performance

Pure water flux of pristine PES, 0.01% CNDs/PES, 0.05% CNDs/PES and 0.5% CNDs/PES composite membranes are demonstrated in Figure 7. At a constant pressure of 300 kPa, the flux of pristine PES, 0.01% CNDs/PES, 0.05% CNDs/PES and 0.5% CNDs/PES was 60.00, 96.93, 142.16 and 196.62 L m^−2^ h^−1^, respectively. The pure water flux increased with increase in CND concentration in the membranes as compared to the pristine PES. This was due to the hydrophilic nature of the CNDs, which is known to increase water flux [30]. Zinadini and Ghalami had similar findings in a study investigating the preparation and characterization of high flux PES nanofiltration membrane using hydrophilic nanoparticles by phase inversion method for application in advanced wastewater treatment [20].

### 3.3. Characterization of Glassy Carbon Electrode (GCE) for Cadmium(II) Determination

The bare GCE was electrochemically characterised using cyclic voltammetry (CV) at a potential range of −0.200 to 0.600 V at a scan rate of 50 mV s^−1^ in in [Fe(CN)_6_]^−3/−4^ redox probe as depicted in Figure 8a. The cadmium(II) peak was observed at 0.200 V. It was observed that for the reversible redox couple, peak current increased with the scan rate at the same potential window. The current signal from the bare GCE is an analytical indicator that the electrode can be used as a suitable platform for electro analysis of cadmium(II) in water.

In this study, the highest stripping current for Cd^+2^ in different electrolytes was obtained by using 0.1 M HCl as an electrolyte (Figure 8b). The supporting electrolyte plays a major role in reducing the internal resistance and electron migration. Since the pH affects the availability of cadmium during stripping, it was varied (1, 2, 4, 5, 6, 8, 10) and pH = 5 was the optimum acidity as displayed in Figure 8c.

The optimisation of the deposition time and potential of Cd^2+^ for stripping on the electrode surface is crucial because it facilitates control of cadmium concentration on the electrode surface to enhance the sensor sensitivity. An electrodeposition potential of −900 mV (Figure 8d) and electrodeposition time of 200 s were chosen as optimized parameters for the pre-concentration step (Figure 8e).

### 3.4. Electrochemical Detection of Cd(II) Using GCE

At optimized conditions (0.1 M HCl, 200 s deposition time and −900 mV deposition potential), GCE electrodes gave an increased current response during the stripping of 10 ppm cadmium at a potential of 0.200 V.

The adsorption of Cd^2+^ from 10 ppm standard solution was done by using five membranes, which were: pure PES membrane, 0.01% CNDs/PES, 0.05% CNDs/PES and 0.5% CNDs/PES. Figure 9 shows that 0.5% CNDs/PES composite was the most efficient membrane composite during Cd^2+^ adsorption because a higher percentage removal of 46.81% was achieved. This proved that 0.5% CNDs embedded on the membrane created more sites for the adsorbate (Cd^2+^) to bind onto the composite membranes. In most studies, where carbon-based nanomaterials are used, higher loadings (above 0.4% or above 0.5%) affected properties such pore size, hydrophilicity, antifouling properties and roughness of the membrane. This is due to the fact that high loadings cause agglomeration of the nanomaterial [31,32].

The electrochemical response during cadmium(II) sensing at different pH values before and after adsorption of Cd^2+^ from 10 ppm standard solution by 0.5 % CNDs on PES membrane was investigated. The adsorption of Cd(II) ions by the 0.5% CNDs/PES membrane was the highest at pH 5 and pH 6, which was 76.08 and 53.62%, respectively as shown in Figure 10. pH studies are mandatory because the ionic states of the analytes depend on the acidity of the surrounding medium [33].

At pH 1, pH 2, pH 4 and pH 8, the % removal of cadmium(II) was 9.67, 6.89, 13.7 and 23.49%, respectively (Figure 10). This was due to the fact that at a strong acidic medium (below pH 3) the concentration of the H^+^ ions on the surface of the membrane is high and there is a competition between these ions and Cd(II), for active sites on the membrane [24]. At this pH the ionic interactions between the carbon nanodots and the cadmium ion is increased. This is because of the dissociation degree of the functional groups (-O-H, -COO- and -C-O-) to form negatively charged species on the membrane surface. This, therefore facilitates electrostatic interaction between the positively charged cadmium ions and the negatively charged oxygen molecules on the membrane surface. Therefore pH 5 was used as the optimum working pH to avoid the precipitation of cadmium as hydroxides at a pH above 6. A similar trend was reported by Tshwenya and Arotiba [34].

The effect of the contact time in removing Cd^2+^ from 10 ppm standard solution was optimised by applying Pure PES, 0.01% CNDs/PES, 0.05% CNDs/PES and 0.5% CNDs/PES membranes for 60 min. The current signals for all membranes decreased for the first 30 min, which meant that adsorption of Cd^2+^ took place within that period (Figure 11). Thereafter, there was a sharp increase from 30 min to 60 min. This trend was as a result of the CND concentration gradient: as the contact time between the composite membrane and Cd(II) solution increased, more Cd(II) ions were able to reach additional adsorption active sites until equilibrium was reached. The 0.5% CNDs/PES membrane composite demonstrated the highest adsorption efficiency compared to pure PES, 0.01% CNDs/PES and 0.05% CNDs/PES composite membranes. This was due to the increase of the CNDs embedded to the PES membrane, which increased the active sites of the composite membrane; 0.5 % CNDs/PES membrane removed most Cd(II) (46.81%) at pH 5 within 30 min of treatment. There was no further increment in percentage removal with the increasing time and this could be due to the saturation of the Cd(II) ions in the membrane. The optimum time was therefore 30 min for this study.

Different standard concentrations of Cd^2+^ (0.5 ppm, 1 ppm, 1.5 ppm, 5 ppm and 10 ppm) were used in determining the percentage adsorbed for Cd(II) ions for 30 min at pH 5 using the different membrane composites. As observed in Figure 12, the percentage Cd(II) adsorbed by pure PES, 0.01 CNDs/PES, 0.05 CNDs/PES and 0.5 CNDs/PES was 95.71, 96.32, 97.69 and 99.78%, respectively.

The concentration of Cd(II) decreased from 10 ppm–0.5 ppm caused a significant percentage of Cd^2+^ to be adsorbed by membranes. The 0.5 ppm cadmium(II) was adsorbed most at pH 5 within 30 min by all the composite membranes. The percentage adsorbed increased with the increase in the percentage of the CNDs added onto the membranes. In Table 4, the removal capacity of cadmium(II) by 0.5% CNDs/PES membranes was 0.70 mg/cm^2^ compared to other membranes and adsorbents.

### 3.5. Interference Studies

The detection of Cd^2+^ could be susceptible to interferences such as the presence of other divalent metal ions. According to the theory, the higher charge of the divalent ions tend to have stronger adsorption strengths compared to monovalent ions [37]. Hence interference studies were carried out at optimised conditions (0.5% CNDs/PES composite membrane, pH 5, 30 min contact time, and 0.5 ppm concentration). The 0.5% CNDs/PES membrane was used to adsorb cadmium(II) in the presence of lead(II), mercury(II) and copper(II) standard solutions. But after the addition of interfering ions, approximately 48% Cd^2+^ removal was achieved.

The current signal of cadmium(II) was mostly suppressed in the presence of copper(II) and lead(II) before the adsorption of cadmium(II) as depicted in Figure 13. After the adsorption, copper(II) is also the most removed metal ion than cadmium(II) because 50% of copper(II) was removed instead of Cd^2+^. Therefore, Cu^2+^ and Pb^2+^ have significant interfering effect during the analysis of Cd^2+^ using GCE in ASV technique. For future work, this membrane can be further optimised to simultaneously remove the analyte and the interfering ions.

### 3.6. Reusability of 0.5% CNDs/PES for Cd(II) Removal

The batch adsorption of cadmium(II) 0.5 ppm standard solution was repeated by using the same membrane (0.5% CNDs/PES) at one-day intervals. After each analysis the membrane was stored in deionised water in the refrigerator and re-used after a day to examine its reusability. Figure 14 displayed that the membrane removed 99.78% Cd^2+^ with relative standard deviation (RSD) of ±3.4% after being analysed on each day. This confirmed that the membrane was consistent during the adsorption of Cd(II) for four days consecutively. After seven days, the membrane started to tear and the % removal decreased from 99.78% to 95.56%.

### 3.7. The Removal of Cadmium(II) in Spiked Water Sample Using 0.5% CNDs/PES

The adsorption behaviour of Cd^2+^ was conducted on a tap water sample spiked with 3ppm of Cd^2+^ as displayed in Figure 15. The concentration detected when using the ASV technique decreased from 3.02 to 1.11 ppm for the spiked water sample, therefore 58.38% of Cd^2+^ was removed.

The comparison of ASV and ICP-OES methods was applied at 95 % confidence level using n = 3 of the spiked water sample. The value of t_critical_ (6.452) was greater than t_observed_ (4.303) for Cd^2+^. This result shows agreement with variation due to random error, and is thus a validation of the reported method.

## 4. Conclusions

The results confirmed that the CNDs were successfully embedded on the PES membrane via phase invasion. TEM displayed well-dispersed and spherical carbon nanodots with uniform shape and an average size of 6.7 ± 2.8 nm. The FTIR confirmed that the combination of PES with CNDs by an introduction of a new functionality at 3400 cm^−1^, which was attributed to -OH due to the presence of CNDs. Raman spectra established that blending the PES with CNDs increased the ID/IG ratio to 0.74, 0.79, 0.83 and 0.82 for Pure CNDs, 0.01% CNDs/PES, 0.05% CNDs/PES and 0.5% CNDs/PES respectively which means that the membrane maintained good structural integrity. SEM demonstrated a relatively spongy and porous surface as the amount of the CNDs was increased from 0.01% to 0.5% in CNDs/PES membranes. AFM results indicated that the surface roughness increased gradually with an increase in the amount of CNDs embedded onto the membranes (0.01% CNDs/PES, 0.05% CNDs/PES and 0.5% CNDs/PES). The higher amount of CNDs coated on the PES reduced the contact angle and improved the hydrophilicity of the membranes. This enhancement in hydrophilicity was attributed to the presence of hydrophilic functional groups such as -OH and COO^−^ found in the CNDs as reported in the FTIR analysis. The membrane performance tested by water flux showed that at a constant pressure of 300 kPa, the flux of pristine PES, 0.01% CNDs/PES, 0.05% CNDs/PES and 0.5% CNDs/PES (60.00, 96.93, 142.16 and 196.62 L m^−2^ h^−1^, respectively) increased proportionally with CNDs concentration in the membranes.

For the detection of cadmium(II) in water, the GCE was electrochemically characterised using CV at a potential range of −0.200 to 0.600 V and a scan rate of 50 mV s^−1^ in [Fe(CN)_6_]^−3/−4^ redox probe. The Cd^2+^ was detected by GCE in ASV under optimised parameters such as 0.1 M HCl (electrolyte), 200 s deposition time and −900 mV electrodeposition potential. The batch adsorption experiments displayed that the 0.5% CNDs/PES removed 99.78% of cadmium(II) from synthetic solutions when pH, time and concentration of Cd^2+^ solution were optimised to 5, 30 min and 0.5 ppm. The interference study showed that Pb^2+^ and Cu^2+^ competed with Cd^2+^ for adsorption on the active sites of the membrane surface, hence lower analyte percentage was detected in their presence. This scenario was also experienced in determination of -Cd^2+^ in spiked water sample, where 58.38% Cd^2+^ was detected due to the sample matrix.

It established that the 0.5% CNDs/PES membrane is reusable because the same membrane was applied for four cycles to remove above 95% of cadmium(II) in water.

## Figures and Tables

**Figure 1 membranes-11-00114-f001:**
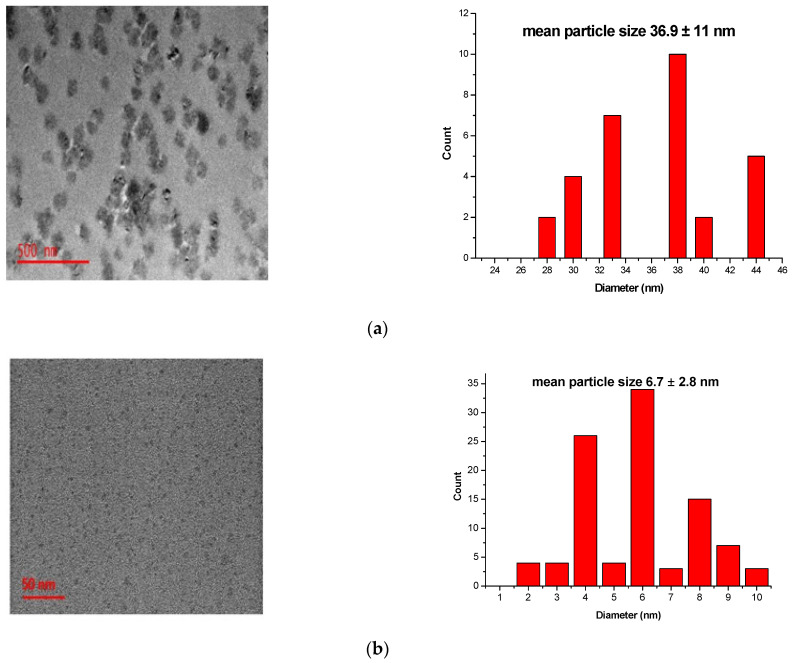
Characterization of the polyethersulfone (PES) and Carbon nanodots coated on polyethersulfone (CNDs/PES) using transmission electron microscopy (TEM). (**a**) before grinding; (**b**) after grinding.

**Figure 2 membranes-11-00114-f002:**
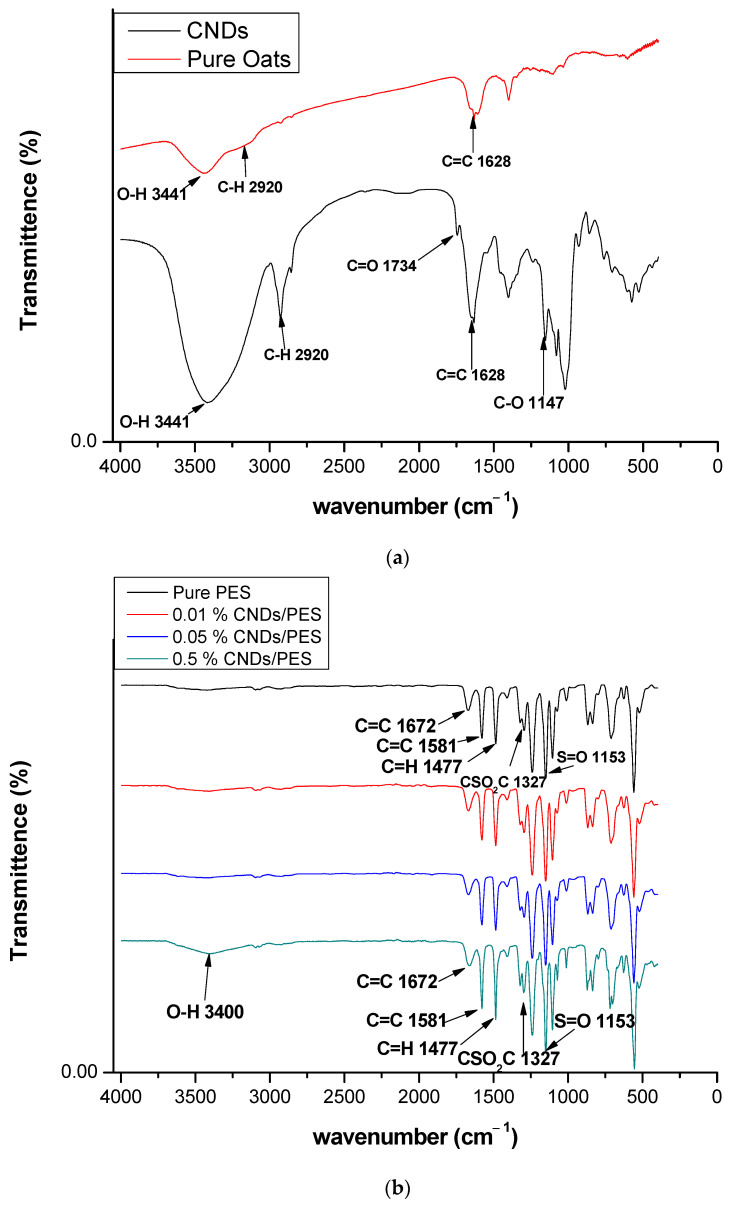
Characterization of the polyethersulfone (PES) and carbon nanodots coated on the polyethersulfone (CNDs/PES) using Fourier transform infrared spectroscopy (FTIR). (**a**) pure oats and CNDs; (**b**) PES and (CNDs/PES).

**Figure 3 membranes-11-00114-f003:**
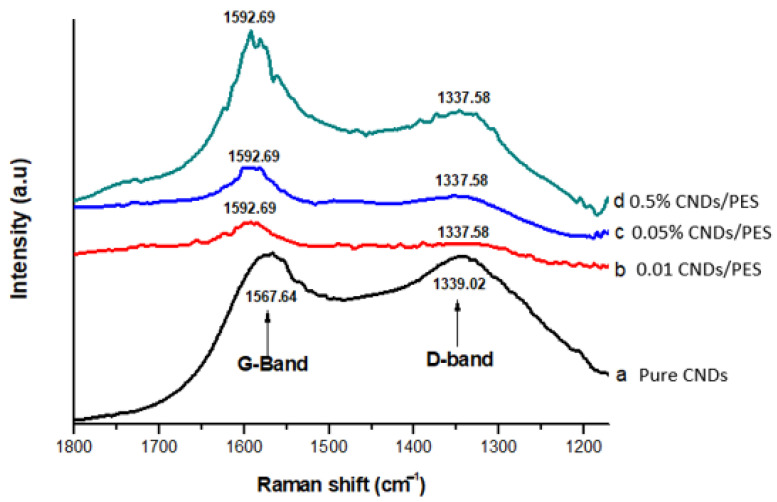
Characterizfation of the polyethersulfone (PES) and carbon nanodots embedded on polyethersulfone (CNDs/PES) using Raman spectra.

**Figure 4 membranes-11-00114-f004:**
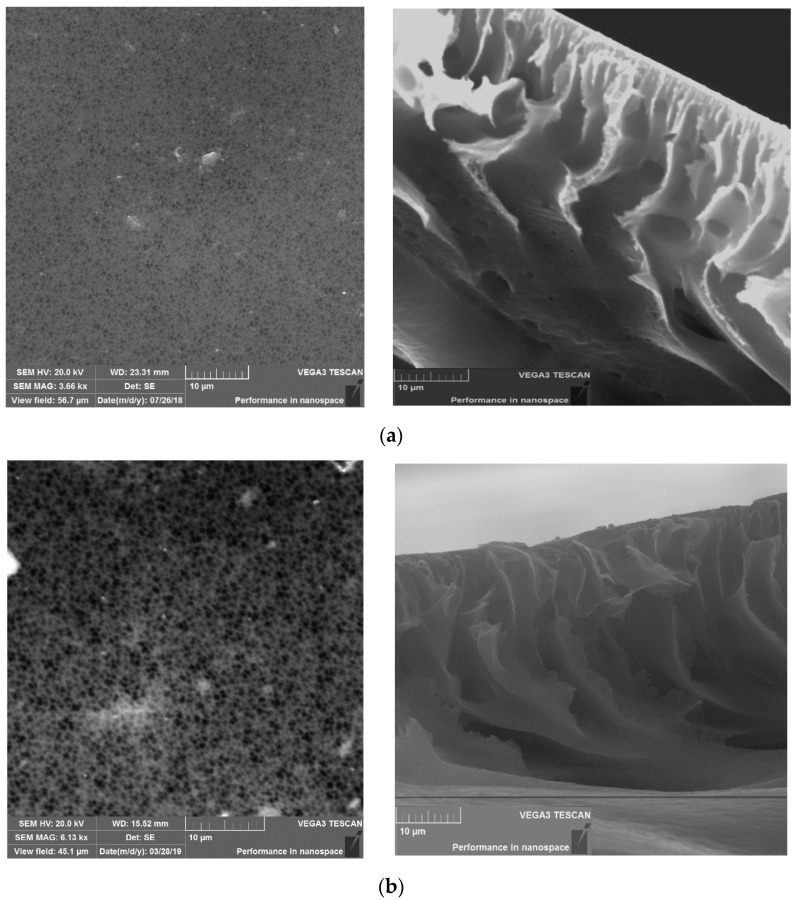
Characterization of the PES and CNDs/PES using scanning electron microscopy (SEM). (**a**) pure PES; (**b**) 0.01% CNDs/PES; (**c**) 0.05% CNDs/PES (**d**) 0.5% CNDs/PES.

**Figure 5 membranes-11-00114-f005:**
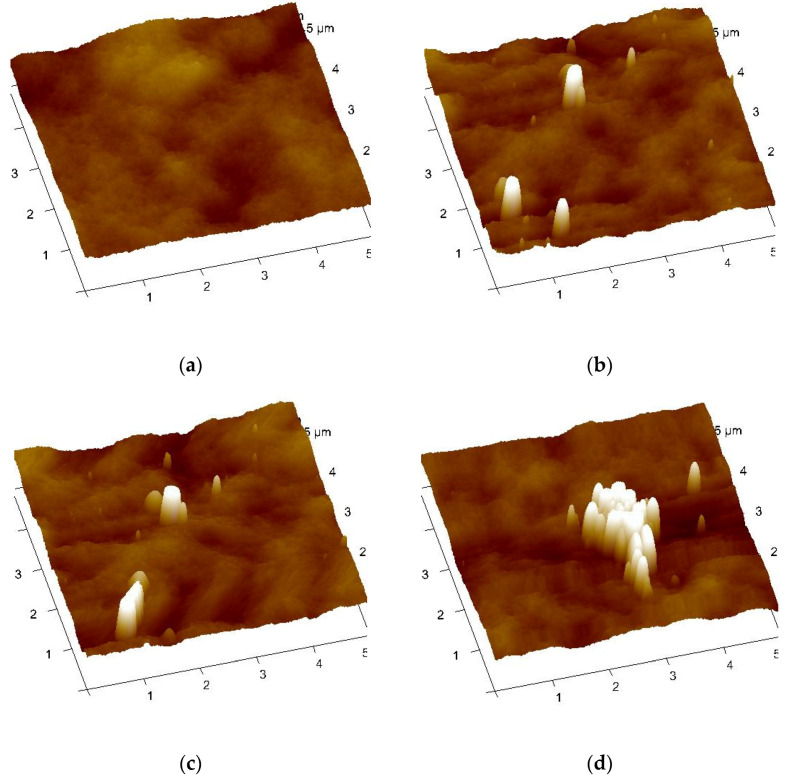
Characterization of the PES and CNDs/PES using atomic force microscopy (AFM). (**a**) pure PES; (**b**) 0.01% CNDs/PES; (**c**) 0.05% CNDs/PES (**d**) 0.5% CNDs/PES.

**Figure 6 membranes-11-00114-f006:**
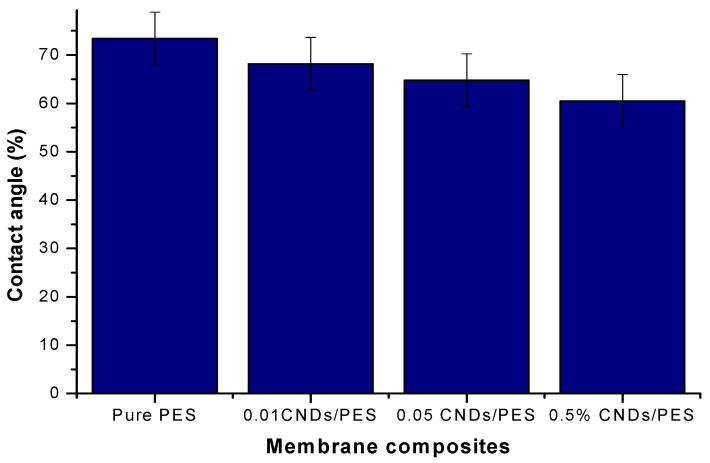
Characterization of the PES and CNDs/PES using Contact angle.

**Figure 7 membranes-11-00114-f007:**
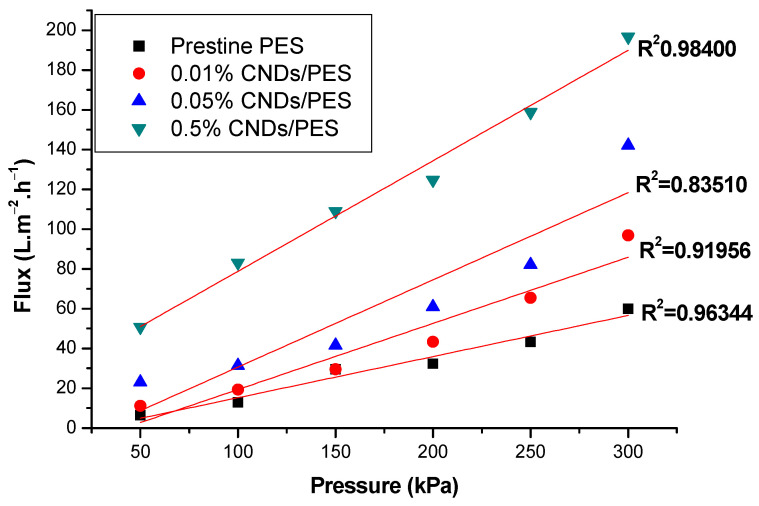
The relationship between pure water flux and pressure in PES and CNDs/PES membranes.

**Figure 8 membranes-11-00114-f008:**
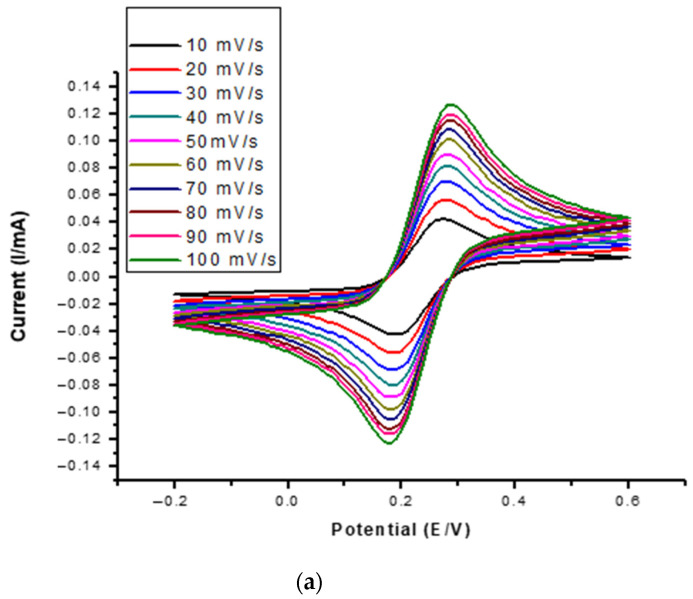
(**a**) Cyclic Voltammetry multi-scan of bare GCE in 10 mM [Fe(CN)_6_]^3−/4−^ redox probe and optimization of: (**b**) electrolytes, (**c**) pH (**d**) deposition time, and (**e**) deposition potential.

**Figure 9 membranes-11-00114-f009:**
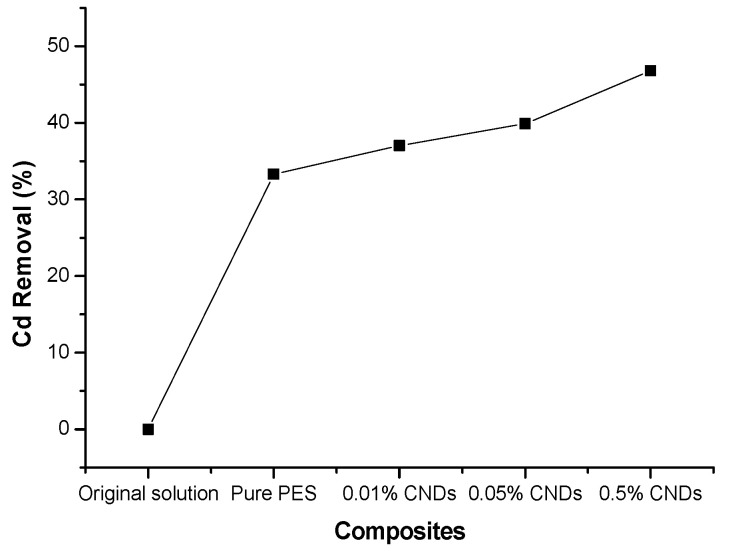
Effect of different membrane composition for the removal of cadmium(II) from standard solutions.

**Figure 10 membranes-11-00114-f010:**
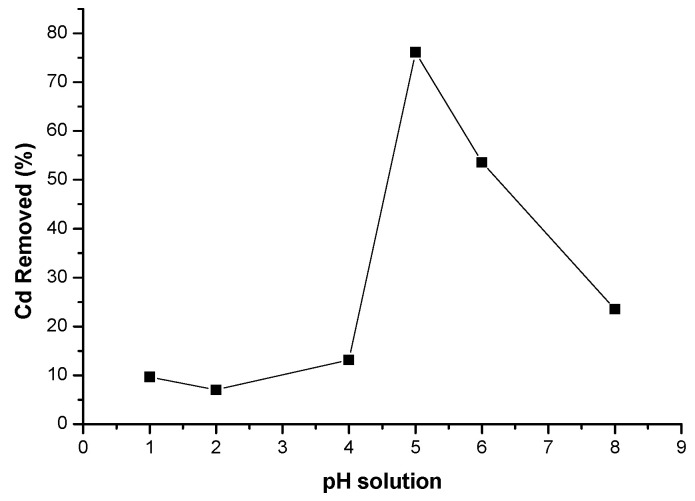
Effect of pH for the removal of cadmium(II) from standard solutions.

**Figure 11 membranes-11-00114-f011:**
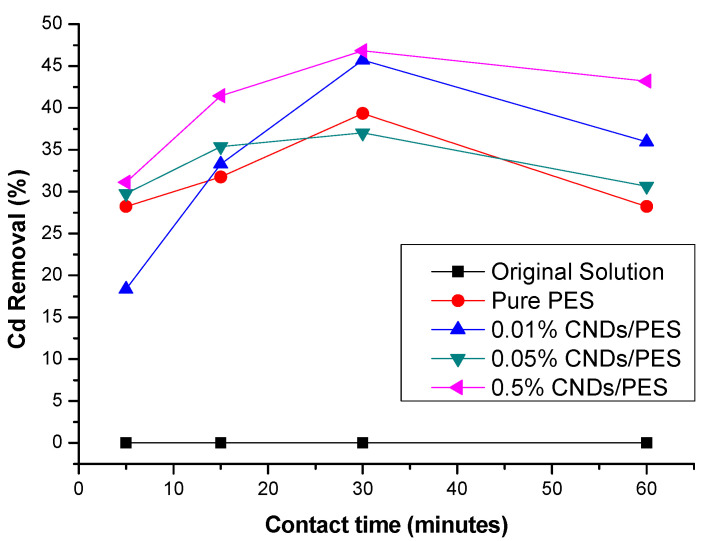
Effect of contact time for the removal of cadmium(II) from standard solutions.

**Figure 12 membranes-11-00114-f012:**
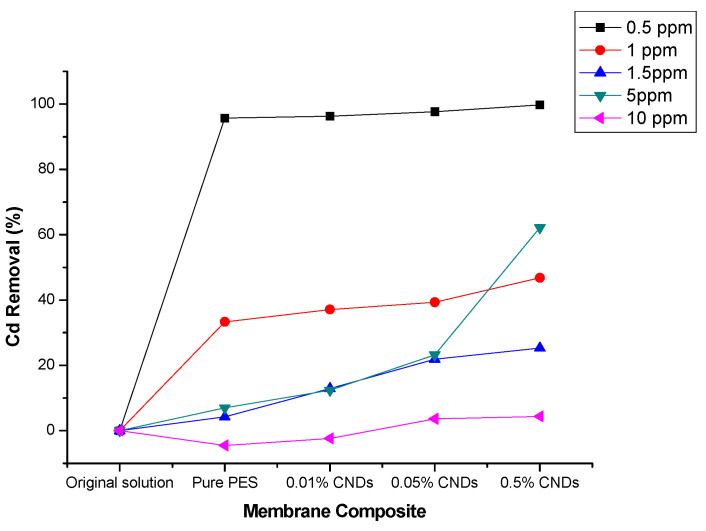
Effect of standard concentrations for the removal of cadmium(II) from standard solutions.

**Figure 13 membranes-11-00114-f013:**
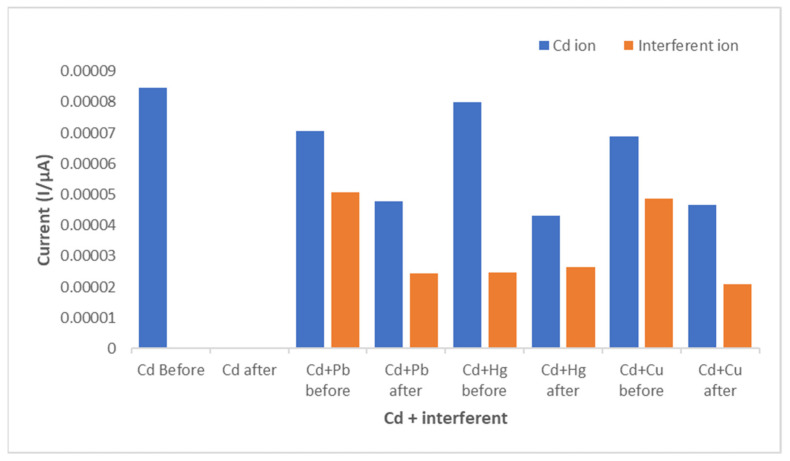
Interference studies during Cd^2+^ detection under optimized conditions.

**Figure 14 membranes-11-00114-f014:**
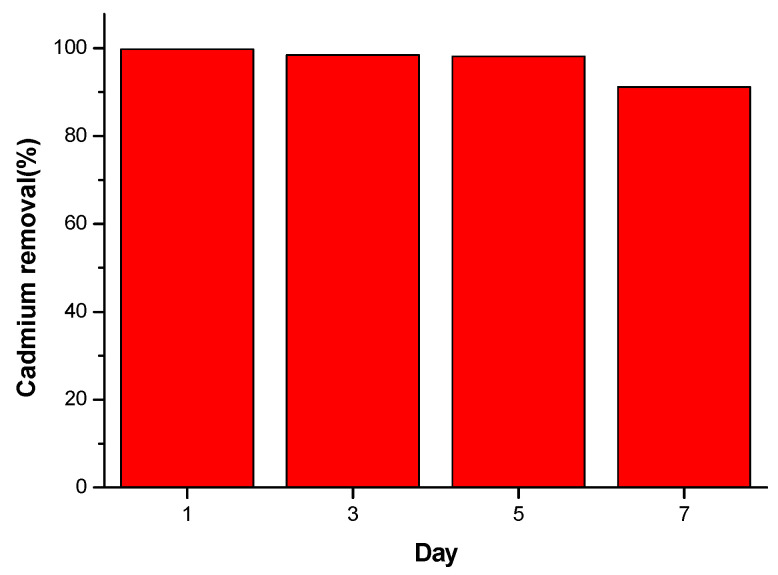
Membrane stability studies during the removal of Cd^2+^ and electrochemical sensing using GCE under optimized conditions.

**Figure 15 membranes-11-00114-f015:**
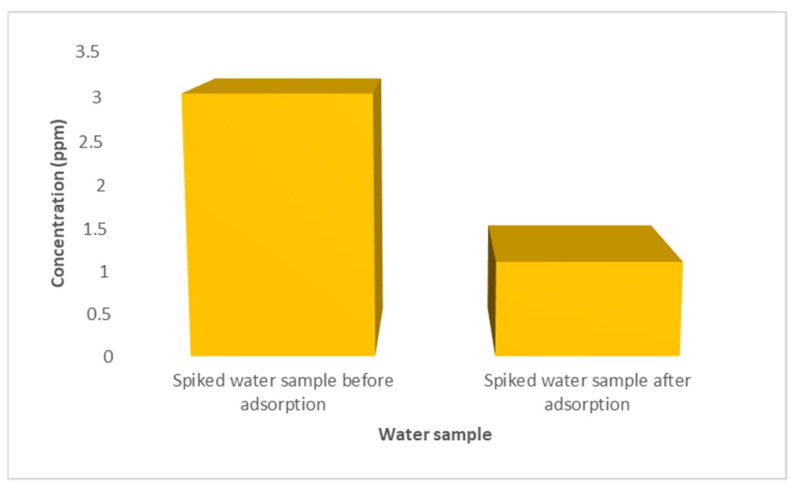
Tap water spiked with 3 ppm cadmium standard.

**Table 1 membranes-11-00114-t001:** Material amounts for composite membrane synthesis.

Membranes	CNDs (g)	NMP (mL)	PES (g)	PVP (g)
Pure PES	0	80	16	2
0.01% CNDs/PES	0.01	80	16	2
0.05% CNDs/PES	0.05	80	16	2
0.5% PES/CNDs	0.5	80	16	2

CNDs: carbon nanodots; NMP: 1-methyl-2-pyrrolidone; PES: polyethersulfone; PVP: polyvinylpyrrollidine.

**Table 2 membranes-11-00114-t002:** I_D_/I_G_ ratio and carbon nanodots embedded membrane composites.

Sample	Peak Position/cm^−1^		I_D_/I_G_
	D-Band	G-Band	
Pure CNDs	1339.02	1567.64	0.74
0.01% CNDs/PES	1337.58	1567.64	0.79
0.05% CNDs/PES	1337.58	1567.64	0.83
0.5% CNDs/PES	1337.58	1567.64	0.82

**Table 3 membranes-11-00114-t003:** AFM statistical data.

Membrane	Surface Roughness (Rq) (nm)
Pristine PES	16.4
0.01% CNDs/PES	21.9
0.05% CNDs/PES	23.7
0.5% CNDs/PES	35.9

**Table 4 membranes-11-00114-t004:** Comparison of differently modified PES membranes for Cd^2+^ removal from standard samples.

Membrane	Detection Technique	% Cd^2+^ Removal Capacity	Literature
Cerium dioxide and composites		93.4 mg/g	[9]
Carbon/zirconium oxide composite	AAS	166.7 mg/g	[10]
Sulfonated magnetic nano-particle adsorbents	AAS	80.9 mg/g	[33]
Emulsion liquid membrane	AAS	0.44 mg/mL and 0.27 mg/g	[35]
A boehmite nanoparticle impregnated electrospun fibre membrane	AAS	0.20–0.21 mg/g	[36]
0.5% CNDs/PES	GCESWASV	0.70 mg/cm^2^	Present work

## Data Availability

The data supporting the study is saved in the Department of Chemical Sciences archives and Simanye Sam google drive. It is available upon request.

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
