# Peer review of "Carbon Nanodots Embedded on a Polyethersulfone Membrane for Cadmium(II) Removal from Water"

_membranes, 2021, doi:10.3390/membranes11020114_

Round 1
Reviewer 1 Report
- The major weakness of this paper is the English language. English needs improvement throughout the manuscript and spelling should be checked thoroughly. There are many spelling mistakes and some phrases are not clear in the original paper. Therefore, for help with English language usage and quality, I strongly recommend that the authors should consult a native English speaker.
- The authors should emphasize the novelty of this system in the introduction. There are a lot of systems for removal of cadmium ions from aqueous solutions. So, their description in the introduction should be added. In addition, some advantages and disadvantages of the PES-based membranes should be also included in the introduction.
- The Figures should be placed in the main close to their discussion. In this way, the reader will follow more easily the results presented in the Figures and the discussion on them.
- The resolution of FTIR spectra should be improved. In addition, the peaks discussed in the main text should be labeled in the FTIR spectra.
- In table 5, the Cd2+ removal capacities should have the same SI units for comparison.
- A characteristic feature of sorbents is their rate of desorption. A discussion on the cadmium ions desorption data should be also added. These studies will offer interesting information on the regeneration of sorbents, which could exhibit several advantages such as: (1) pollutant recovery, (2) reusability of sorbent, (3) reduction of the process cost, (4) decrease of the resulted waste (if the sorbent cannot be recovered, this will constitute a major drawback), and (5) identification of the sorption mechanism.
- The reusability of the CNDs/ polyethersulfone membranes for removal of cadmium ions through several cycles of adsorption/desorption is crucial in practical applications. Therefore, the possibility of recycling the the CNDs/polyethersulfone should be also investigated and the results must be included in the revised manuscript.
Author Response
Dear Reviewers
Thank you for your positive contributions to our work.
We hope that we addressed your concerns in the manuscript.
Kind regards

Reviewer 2 Report
This paper prepares CND/PES composite membranes for Cadmium adsorption. CND is produced in lab from oats. Different amount of CND is loaded in the PES membranes. Successful attainment of the membranes are confirmed by various techniques, such as FTIR, Ramen and SEM. The incorporation of CND changes the morphology and hydrophilicity of the membranes. These changes bring about enhancement in membrane flux and Cadmium adsorption. The paper is well prepared, and is recommended for publication after minor revision.
- 0.5% CND/PES membrane shows a good performance and intact structure. Can the loading amount be higher?
- From SEM images, the surface becomes more porous and the cross-section contains less macrovoids. These changes are important to membrane separation and adsorption properties. More explanation on morphology should be provided.
- What is the adsorption rate for monovalent ions? Is there any competitive advantage towards Cadmium?
Author Response
Dear Reviewers
Thank you for your constructive comments.
We hope that we have addressed them well.
Kind regards

Reviewer 3 Report
In this study, green carbon nano- 9 dots were synthesised from oats and embedded on polyethersulfone membrane via phase inversion 10 for the adsorption of Cd2+ from water. Characterisation techniques for the CNDs and PES mem- 11 branes were TEM, FTIR, SEM, Raman spectroscopy, AFM, contact angle and pure water flux assess- 12 ment system operated at 300 kPa. TEM analysis confirmed that the CNDs were well dispersed with 13 uniform shape and size (6.7±2.8 nm). Raman analysis illustrated that the CNDs were embedded on 14 the PES and the ID/IG ratio slightly increased showing that the membranes maintained good struc- 15 tural integrity. The CNDs/PES improved hydrophilicity compared to PES. ASV technique detected 16 99.78% Cd2+ removal by 0.5% CNDs/PES at optimum conditions: 30 min. contact time, at pH 5 and 17 0.5 ppm Cd2+ solution. I recommend the paper for major revision prior to a decision. In view of that, comments are appended below:
- The English language need to check carefully in the revision stage because of careless mistake in several positions. In addition, the manuscript should be thoroughly checked for English corrections as there are some colloquial terms being used.
- The optimum concentration for Cd2+ solution was 5 ppm in this study. This concentration comes under permissible limit for water intake or not?
- Include FTIR, SEM, XPS after Cd(II) removal also and explain the changes.
- How the removal of Cd using membrane is comparable with conventional adsorbents?
- Explain the abbreviated words when they are using first time.
- Improve the quality of all figures.
- Abstract: This section is completely different than the Introduction and Experimental sections. The main findings with important opinions are acceptable. The mathematical terms need to be added. The authors need to consider these points in the revision stage.
- I suggest that the authors can adapt the conclusions in accordance with the proposed objectives and revealing the importance of the title employed.
- Introduction part must be updated. Authors may discuss about some more toxic metal ions because several other metals also exist in the environment with Cd(II). Several materials have been already used for the removal of metal ions which may be discussed here in brief. As a suggestion, the authors may follow the following work or other similar work for the discussion in the introduction part to make it interesting at broad level: Environmental Chemistry Letters, 16 (2018) 1233–1246; Carbon letters 8 (3), 199-206, 2007; Journal of Water Process Engineering 33, 101112, 2020; Journal of Molecular Liquids 310, 113025, 2020; Environmental Chemistry Letters 16 (4), 1339-1359, 2018
- Result and discussion section (including materials characterization) must explain with more experimental findings, such as experimental results, significance of work, finding of results and choice of materials.
Author Response
Dear Reviewer
Thank your positive contribution into our work.
We hope that we have addressed your concerns.
Kind regards

Round 2
Reviewer 1 Report
The authors have followed all recommendations in the revision of their manuscript and therefore the manuscript could be accepted in this form.
Reviewer 3 Report
Still, the quality of figures need the improvement.